Galectin-3 promotes fibrosis in ovarian endometriosis

Yang Guimin
Deng Yupeng
Cao Guangming
Liu Chongdong Liuchongdong@ccmu.edu.cn
Department of Obstetrics and Gynecology, Beijing Chaoyang Hospital of Capital Medical University , Beijing , China
Shi Huashan
Electronic publication date: 2024 Feb 14
Publication date: 2024
Volume: 12
Electronic Location ID: e16922
Received 2023 Oct 18; Accepted 2024 Jan 19
Copyright: ©2024 Yang et al.
Copyright year: 2024
Copyright holder: Yang et al.
License: This is an open access article distributed under the terms of the Creative Commons Attribution License, which permits unrestricted use, distribution, reproduction and adaptation in any medium and for any purpose provided that it is properly attributed. For attribution, the original author(s), title, publication source (PeerJ) and either DOI or URL of the article must be cited.
License URL: https://creativecommons.org/licenses/by/4.0/

Keywords: Ovarian endometriosis, Fibrosis, Galectin-3, 12Z cells

Funding: Key Special Project of the “Ministry of Science and Technology on the Prevention and Control of Reproductive Health and Major Birth Defects” of the China No. 2017YFC1001204 This study was supported by the Key Special Project of the “Ministry of Science and Technology on the Prevention and Control of Reproductive Health and Major Birth Defects” of the China (No. 2017YFC1001204). The funders had no role in study design, data collection and analysis, decision to publish, or preparation of the manuscript.

==============================
Objective

This study aimed to investigate the potential role of galectin-3 (Gal-3) in the pathogenesis of fibrotic alterations in ovarian endometriosis (OVE).

Methods

In this study, we collected the ectopic endometrial tissues and eutopic endometrial tissues from 31 OVE patients treated by laparoscopy, and the eutopic endometrial tissues from 23 non-OVE patients with leiomyoma or other benign diseases were used as control. Hematoxylin and eosin (H&E) and Masson’s trichrome staining were utilized for histopathological assessment. The primary normal endometrial stromal cells (NESC), ectopic endometrial stromal cells (ECSC), and eutopic endometrial stromal cells (EUSC) were isolated. Gal-3 overexpression plasmids (Gal-OE) and short hairpin RNA targeting Gal-3 (Gal-3-shRNA) were transfected into the immortalized human endometriotic cell line 12Z, respectively. RT-qPCR, Western blot analysis, and immunohistochemistry were used to detect the mRNA and protein expression levels of Gal-3, type I collagen (COL-1), connective tissue growth factor (CTGF) and α-smooth muscle actin (α-SMA), respectively.

Results

H&E and Masson staining showed that ovarian ectopic endometrium exhibited glandular hyperplasia, high columnar glandular epithelium, apical plasma secretion, more subnuclear vacuoles, and obvious fibrosis, compared with normal endometrium. The mRNA and protein levels of Gal-3 , CTGF, α-SMA, and COL-1 were all upregulated in the ectopic endometrial tissues of OVE patients compared to the eutopic endometrial tissues from OVE patients and non-OVE patients. Moreover, ECSC expressed higher levels of Gal-3, CTGF, α-SMA, and COL-1 than EUSC and NESC. Follow-up investigations demonstrated that the Gal-3 overexpression substantially increased fibrosis-related markers including CTGF, α-SMA, and COL-1 within the 12Z cell line. Conversely, Gal-3 knockdown showed the opposite effects.

Conclusion

Gal-3 promotes fibrosis in OVE, positioning it as a prospective therapeutic target for mitigating fibrosis in endometriosis.

Introduction

Endometriosis is a widespread gynecological condition that impacts between 6% and 10% of women in the reproductive-age worldwide. This condition is distinguished by the atypical proliferation and implantation of endometrial tissue outside the cavity of the uterus, frequently influencing the ovaries and peritoneum (Nanda et al., 2020). The condition is clinically stratified into three subtypes: ovarian endometriosis (OVE), deep infiltrating endometriosis (DIE), and peritoneal endometriosis (PE) (Qiu et al., 2023; Rolla, 2019). In particular, OVE is the most common lesion in endometriosis that is frequently detected by gynecologists on ultrasonography (Collins et al., 2019). However, the pathogenesis of endometriosis is not clear.

It has been reported that tissue fibrosis caused by repeated tissue injury and repair is one of the primary pathological mechanisms of endometriosis (Ganieva et al., 2020). Fibrosis is defined by the localized expansion and differentiation of fibroblasts, along with the anomalous buildup of an excessive collagen matrix, histopathologically (Matsuzaki, Pouly & Canis, 2020). During the pathogenesis and progression of endometriosis, extensive extracellular matrix remodeling and deposition occur, histologically manifesting as an overabundance of dense fibrous tissue encapsulating endometrial glands and stroma (Wu et al., 2018). The primary cell types including ectopic endometrial cells, macrophages and platelets were involved in fibrosis formation, and cell interactions among each cell types lead to the development of fibrosis (Viganò et al., 2020). Fibrosis is always present in all subtypes of endometriosis (Rolla, 2019). Excessive fibrosis in endometriosis not only causes scar formation, dysmenorrhea, dyspareunia, and chronic pelvic pain, but also affects the egg-collection function of the fallopian tube and reduces the reserve of follicles, resulting in infertility in patients with endometriosis (Garcia Garcia et al., 2023). DIE is an estrogen-dependent disease and accounts for approximately 10%–15% of endometriosis cases, but it usually exhibits resistance hormone suppression therapy due to excess fibrosis (Mariani et al., 2021). The most favorable long-term prognostic results and substantial relief from symptoms are achieved through optimal surgical removal of the endometriotic lesions (Ota, Andou & Ota, 2018; Chiu et al., 2022). The loss of ovarian parenchyma during surgery and postoperative follicular reserve are related to the diameter and the degree of fibrosis of ovarian endometriosis cyst. However, to the best of our current understanding, the processes and underlying mechanisms of fibrogenesis in endometriosis remain largely enigmatic. Consequently, elucidating the mechanistic pathways driving fibrosis in endometriosis, particularly within the context of ovarian endometriomas, is of paramount importance.

Galectin-3 (Gal-3) belongs to the galactose-binding lectin family, which is unique for its specific binding affinity to β-galactoside units and has at least one evolutionarily conserved carbohydrate recognition domain (Dong et al., 2018). Out of the 15 galectins found in mammalian systems thus far, Gal-3 is notable for being the sole chimeric galectin. Intracellularly, Gal-3 is predominantly localized in the cytoplasm but is also capable of nuclear translocation. Additionally, it can be secreted onto the cell surface and released into various body fluids, including serum and urine (Nangia-Makker, Hogan & Raz, 2018). Prior investigations suggested that increased Gal-3 expression is linked to the remodeling of cardiovascular structures, leading to cardiac fibrosis and eventual heart failure (Blanda et al., 2020). Moreover, Gal-3 was identified as a key regulator in lung fibrosis, and Gal-3 inhibition may be an effective treatment method for pulmonary fibrosis (Fulton et al., 2019). Twist1 regulates macrophage plasticity to promote renal fibrosis through galectin-3. It has been shown that Twist1/Gal-3 signaling facilitates renal fibrosis through regulating M2 macrophage polarization (Wu et al., 2022). Therefore, Gal-3 may act as diagnostic or prognostic indicators in fibrotic disorders affecting the heart, renal system, and pulmonary tissues. Notably, Gal-3 expression is increased after persistent tissue injury, causing chronic inflammation, accompanied by fiber and scar formation, isolating the injury tissues from the surrounding healthy tissues (Wang et al., 2023). However, the effects of Gal-3 on the fibrosis in endometriosis has not been elucidated.

Connective tissue growth factor (CTGF), type I collagen (COL-1), and α-smooth muscle actin (α-SMA), are key indicators of fibrosis. These markers play a crucial role in the fibrotic processes associated with endometriosis (Toda et al., 2018; Dolivo, Weathers & Dominko, 2021). Given the roles of Gal-3 in fibrotic disorders, we speculated that Gal-3 may affect fibrosis development in endometriosis. In this study, we investigated the expression level of Gal-3 in ectopic endometrial tissues of OVE patients, and detected the effects of Gal-3 on fibrosis-related indicators including α-SMA, CTGF, and COL-1. We aimed to clarify the effects of Gal-3 on fibrosis in endometriosis development and provide a better understanding of the pathogenesis of fibrosis in endometriosis and opens avenues for future research in this field.

Materials and Methods

Collection of tissue samples

This research received ethical approval from the Ethics Committee of Beijing Chaoyang Hospital of Capital Medical University and abided by the ethical guidelines of the Declaration of Helsinki (2016-166). All subjects participated in the study gave their informed consent by signing the appropriate documents. We recruited a well-defined cohort of 31 OVE patients, aged between 20 and 45 years, who underwent laparoscopic intervention at Beijing Chaoyang Hospital affiliated with Capital Medical University. The inclusion criteria were as follows: (1) OVE diagnosis confirmed by histopathology. (2) Patients with comprehensive clinical data and available tissue samples. (3) Patients were not received any medications and treatments. Patients with adenomyosis and endometrial polyps, malignant tumors, and severe organ diseases were excluded. Moreover, 23 patients (20–42 years) with leiomyoma or other benign diseases without endometriosis were used as controls. All patients with hysteromyoma were intramural and or subserosal myoma. Submucosal myoma and adenomyoma were excluded. All patients had not received hormone therapy and intrauterine device placement within 6 months before the operation. Every patient in the study had regular menstrual cycles, varying between 26 and 32 days. Ectopic endometrial tissues and matching eutopic endometrial tissues were collected from OVE patients. Normal eutopic endometrial tissues were also gathered from non-OVE patients. Tissue samples were immediately frozen in liquid nitrogen and stored at −80 °C until use.

Hematoxylin and eosin (H&E) staining

Tissue specimens were subjected to fixation using a 10% phosphate-buffered formalin solution for a duration of 24 h. Following the fixation, the samples underwent standard histological processing that encompassed dehydration, clearing, and infiltration, culminating in their embedding in paraffin blocks. Then, tissue sections were carefully acquired using a microtome to get 5 µM thickness of each section. To visualize the nuclear and cytoplasmic components, the sections were subsequently stained using the H&E technique. Histological analysis was conducted employing a high-resolution optical microscope (OLYMPUSBX41; Olympus, Tokyo, Japan). At a magnification of 100 ×, photomicrographs were taken to facilitate a thorough morphological study.

Masson staining

Before being embedded in paraffin, fixation of the tissue pieces was done in 4% paraformaldehyde. Following fixation, standard 4 µM tissue slices were subjected to Masson staining using a kit (Wuhan Bode Bioengineering Co., Ltd., Wuhan, China). Histological assessments were carried out via an OLYMPUS BX41 optical microscope (Olympus, Tokyo, Japan), with images documented at 100 × magnification. Increased blue coloration indicated elevated levels of collagen and fibrotic tissue formation.

Primary cell isolation and culture

Within a sterile surgical setting, tissue samples ranging from 3–5 mm in diameter were collected in an aseptic manner from ectopic and eutopic endometrial areas in patients diagnosed with OVE. Eutopic endometrial tissues from patients without OVE were also aseptically gathered for comparison. These tissue samples were individually placed into 15 ml centrifuge tubes containing serum-free DMEM/F-12 medium (Gibco). To commence the enzymatic digestion process, a solution consisting of 2 ml of 0.1% type I collagenase (Sangon Biotech Co., Ltd., Shanghai, China) was introduced. The resulting mixture was subjected to shaking and digestion at a temperature of 37 °C for a period of 1 h. The process of digestion was halted by the addition of complete DMEM/F-12 medium. Furthermore, the mixture underwent filtration, and the resulting filtrate was centrifuged (800 rpm for 6 min). The pellet obtained was resuspended in DMEM/F-12 complete media and incubated at 37 °C.

Primary cell identification

To evaluate the purity and consistency of eutopic endometrial stromal cells (EUSC), normal endometrial stromal cells (NESC), and ectopic endometrial stromal cells (ECSC), immunofluorescence staining was utilized with anti-vimentin and anti-cytokeratin antibodies. In a 24-well plate equipped with coverslips, 8 ×104 cells/well were seeded. Upon reaching a 30–50% confluence, the cells were subjected to fixation with a 4% paraformaldehyde solution for 20 min. The coverslips were then blocked at ambient temperature for half an hour, followed by incubation with primary antibodies at a temperature of 4 °C overnight. Afterward, secondary antibodies (Abcam Technology, Cambridge, UK) were incubated for an hour at ambient temperature. Following extensive washing, cells were arranged on slides, and DAPI (Abcam Technology, Cambridge, UK) was applied. Lastly, fluorescent microscopy (Olympus, Tokyo, Japan) was employed for immediate observation and imaging.

Cell transfection

The 12Z human endometriotic cell line was obtained from ATCC (Manassas, VA, USA), cultured in DMEM/F-12 medium enriched with FBS (10%), and incubated at 37 °C (5% CO2). For genetic experiments, Gal-3 overexpression vector (Gal-3-OE), short hairpin RNAs (shRNAs) targeting Gal-3, and their respective negative controls, were designed and produced by GenePharma (Shanghai, China). Specifically, to generate Gal-3-OE, DNA fragment containing Gal-3 sequence and pcDNA.3.1 vector underwent double digestion using restriction endonucleases BamH I and Age I. Following this, the two products were joined using T4-DNA ligase, followed by transformed into competent cells. Monoclonal colonies were chosen for cultivation and positive transformants were screened. The constructed vector was verified by double digestion and sequencing analysis. Sequencing analysis and double digestion were used to validate the generated vector. Empty pcDNA.3.1 vector was used as its negative control (NC-OE). To construct shRNAs targeting Gal-3, we designed three shRNAs containing different target sequences (Gal-3-shRNA-1, Gal-3-shRNA-2, Gal-3-shRNA-3). Gal-3-shRNA-1 (5′-CCC ACG CTT CAA TGA GAA CAA-3′),Gal-3-shRNA-2 (5′-CCC ACG CTT CAA TGA GAA C-3′), Gal-3-shRNA-3 (5′-GCA AAC AGA ATT GCT TTA GAT-3′), and negative control NC-shRNA (5′-GCA AAC AGA ATT GCT TTA G-3′). These constructs were introduced into cells by transfection using Lipofectamine 3000 (Invitrogen, Waltham, MA, USA). Briefly, 10 µL of Lipofectamine 3000 reagent and the designed constructs were respectively diluted with 250 µL of Opti-MEM solution. Following a 5-minute standing period at room temperature, the two mixed solutions were mixed and stood at room temperature for 20 min. Thereafter, the mixture was introduced into the 6-well culture plate upon the cell confluence had reached 70–80%, and incubated for a duration of 48 h.

RT-qPCR

The tissue samples and cells were subjected to total RNA extraction using Trizol reagent (Tiangen Biotech Co., Ltd., Beijing, China). The subsequent cDNA was synthesized by the TIANScript cDNA kit (Tiangen Biotech Co., Ltd., Beijing, China). RT-qPCR was performed using an SYBR Green Master Mix equipment (Bio-Rad, Hercules, CA, USA) in combination with ABI 7500 System (Applied Biosysterns, USA). Specific primers for the target genes were as follows. Gal-3 (93 bp): 5′-CTT CCA CTT TAA CCC ACG CTT CAA-3′ (forward), 5′-TGT CTT TCT TCC CTT CCC CAG TTA TT-3′ (reverse); α-SMA (177 bp): 5′-GGA GCA TCC GAC CTT GCT AA-3′ (forward), 5′-CCA TCT CCA GAG TCC AGC AC-3′ (reverse); COL-1 (167 bp): 5′-CAG TGG CGG TTA TGA CTT CAG-3′ (forward), 5′-GGC TGC GGA TGT TCT CAA TC-3′ (reverse); CTGF (147 bp): 5′-TAG CTG CCT ACC GAC TGG AA-3′ (forward), 5 ′-CTT AGA ACA GGC GCT CCA CT-3′ (reverse); GAPDH (131 bp): 5′-ATG ACC CCT TCA TTG ACC-3 ′ (forward), 5′-GAA GAT GGT GAT GGG ATT TC-3′ (reverse). The relative levels of mRNA expression were calculated by the 2−ΔΔCT method. GAPDH served as an internal reference gene for normalization.

Western blot analysis

The total proteins were isolated using ice-cold RIPA lysis buffer obtained from the Beyotime Institute of Biotechnology in Shanghai, China. The homogenate was centrifuged (12,000 rpm for 15 min) at 4 °C. After centrifugation, the supernatant was collected cautiously. Next, the modified BCA Protein Assay Kit (Beyotime Institute of Biotechnology, Shanghai, China) was used to measure the quality of proteins in the supernatants. Aliquots of equal volume, each containing 40 µg of protein, were separated using sodium dodecyl-sulfate polyacrylamide gel electrophoresis (SDS-PAGE). Electrotransfer was then used to transfer the protein bands onto nitrocellulose membranes (Pall Company, New York, United States). Subsequently, these nitrocellulose membranes were laid onto polyvinylidene fluoride (PVDF) membranes. The membranes were blocked with non-fat powdered milk (Sangon Biotechnology, Shanghai, China) at room temperature for one hour. To facilitate antigen detection, the membranes were subjected to incubation with primary antibodies for overnight incubation (4 °C). Subsequently, at room temperature (1 h) the suitable horseradish peroxidase-conjugated secondary antibody was added. The immunodetection process was done by employing the enhanced chemiluminescence (ECL) western blotting substrate reagent manufactured by Tsea bioscience Co., located in Shanghai, China. The immunoblots were subjected to quantitative analysis utilizing the JS-780 Automatic Gel Imaging Analysis Systems and Molecular Imager ChemiDoc XRS (BIO-RAD Co., California, USA). The antibodies purchased from Abcam (Cambridge, UK) are listed as follows: Gal-3 (1:5000, ab76245), COL-1 (1:1000, ab34710), CTGF (1:500, ab5097), α-SMA (1:1000, ab5694), GAPDH (1:2500, ab9485), and secondary antibody (1:2000, ab6721).

Immunohistochemistry

The harvested tissues were initially fixed in a 4% formaldehyde solution overnight, subsequently undergoing a dehydration process in a 30% sucrose solution maintained at 4 °C. Next, 6 µm frozen sections were made continuously. The tissue slices were submerged in PBS buffer and rewarmed at 4 °C for 15 min. Subsequently, the blocking of samples was performed utilizing a 5% bovine serum albumin (BSA) solution at a temperature of 37 °C for 30 min. The samples in the presence of the primary antibodies (anti-Gal-3, anti- α-SMA, anti-CTGF, and anti-COL-1 antibodies obtained from Abcam (Cambridge, UK) were incubated at a temperature of 4 °C overnight. Following the rinsing of the tissue samples with PBS, they were subjected to a 30-minute incubation at 37 °C with a secondary antibody, which was horseradish peroxidase (HRP)-labeled sheep anti-mouse secondary antibody. Following three successive 5-minute washes with PBS, the color reaction was executed utilizing a diaminobenzidine (DAB) solution. The image was observed under the microscope. Following the DAB reaction, sections were subjected to hematoxylin staining at room temperature for a period of 1 min. After a thorough rinse with PBS, neutral gum was used to seal the sections. After that, they were examined and captured on camera with an optical microscope. Result criteria: Cells with positive protein expression were stained with sepia or tan in the cell membrane and cytoplasm.

Statistical analysis

The statistical analyses were conducted using version 26.0 of IBM SPSS Statistics. Data from three independent experiments were displayed as means ± standard deviation (SD). The two groups were compared with an unpaired Student’s t-test. The one-way analysis of variance (ANOVA) followed by least significant difference (LSD) post-hoc t-tests was done conducting for pairwise comparisons between groups. A p < 0.05 was regarded as statistically significant.

Results

Pathological changes of H&E and masson staining

The HE staining was shown in Fig. 1A. Under the microscope, the eutopic endometrial tissue showed normal and regular gland morphology. Glandular epithelial cells are columnar, arranged in order, with abundant cytoplasm and round or oval nucleus. Whereas ectopic endometrial tissue showed glandular hyperplasia. Glandular epithelial cells were highly columnar, monolayer or pseudostratified, with mostly oval nucleus. Apical plasma secretion and many subnuclear vacuoles can be observed near the base. Masson staining (Fig. 1B) showed that there was no obvious fibrosis in the normal endometrial tissues, and the degree of fibrosis of ectopic endometrial tissues was higher than that of eutopic endometrial tissues in OVE patients. Our results indicated that fibrosis may contribute to the development of OVE.

Figure 1 The images of H&E staining and Masson staining.

(A) H&E staining; (B) Masson staining. The images captured at a magnification of 100× serve as representative visual representations of three distinct and separate samples. OVE, ovarian endometriosis.

The protein and mRNA expression levels of Gal-3, COL-1, CTGF and α-SMA in endometrial tissues

To elucidate the role of fibrosis in the development of OVE, we determined the expression of fibrosis-related genes in clinical tissue samples. RT-qPCR results demonstrated that the mRNA expression levels of Gal-3 (Fig. 2A), CTGF (Fig. 2B), α-SMA (Fig. 2C), and COL-1 (Fig. 2D) were all significantly increased in the ectopic endometrial tissues of OVE patients compared to the eutopic endometrial tissues of OVE patients (p < 0.01) and non-OVE control subjects (p < 0.01). Likewise, Western blot results further corroborated that the protein levels of Gal-3, CTGF, α-SMA, and COL-1 were all enhanced in the ectopic endometrial tissues of OVE patients compared with eutopic endometrial tissues of OVE patients (p < 0.01) and non-OVE control subjects (p < 0.01) (Figs. 3A–3E). Immunohistochemical results were in concordance with the Western blot results, confirming the elevated levels of protein expression of Gal-3, COL-1, α-SMA, and CTGF in ectopic endometrial tissues (p < 0.01) (Figs. 4A–4E).

Figure 2 The mRNA expression levels of Gal-3, COL-1, CTGF, and α-SMA in clinical endometrial tissues.

The levels of mRNA expression of (A) Gal-3, (B) COL-1, (C) CTGF, and (D) α-SMA detected by qRT-PCR in tissue samples. Data were presented as mean ± SD. * p < 00.05, ** p < 00.01. OVE, ovarian endometriosis.

Figure 3 The protein expression levels of Gal-3, COL-1, CTGF, and α-SMA in clinical endometrial tissues.

(A) The levels of protein expression of (B) Gal-3, (C) COL-1, (D) CTGF, and (E) α-SMA were detected by Western blot analysis in tissue samples. Data were presented as mean ± SD. * p < 00.05, ** p < 00.01. OVE, ovarian endometriosis.

Figure 4 The protein expression levels of Gal-3, COL-1, CTGF, and α-SMA in clinical endometrial tissues.

(A) The levels of protein expression of (B) Gal-3, (C) COL-1, (D) CTGF, and (E) α-SMA in tissue samples were shown by immunohistochemistry. Data were presented as mean ± SD. * p < 00.05, ** p < 00.01. OVE, ovarian endometriosis.

Primary cell identification results

Immunofluorescence staining was executed to identify the isolated primary endometrial stromal cells. As evidenced in Fig. 5, these cells exhibited positive immunoreactivity for vimentin protein but not keratin 7 protein. The results substantiated that endometrial stromal cells were successful isolated and the proportion was more than 95%.

Figure 5 The images of immunofluorescence staining of primary cells.

The provided images, magnified at 100 ×, serve as representative images of three distinct and separate samples. NESC, normal endometrial stromal cells; ECSC, ectopic endometrial stromal cells; EUSC, eutopic endometrial stromal cells.

The protein and mRNA expression levels of Gal-3, COL-1, CTGF and α-SMA in endometrial stromal cells

Subsequently, we assessed the expression profiles of Gal-3, COL-1, CTGF, and α-SMA in the endometrial stromal cells. RT-qPCR data illustrated a significant upregulation of Gal-3 COL-1, CTGF, and α-SMA mRNA expression in ECSC in comparison to NESC (p < 0.01) and EUSC (p < 0.01) (Figs. 6A–6E). Similarly, Western blot results demonstrated that the protein levels of Gal-3, COL-1, CTGF, and α-SMA were substantially increased in ECSC compared to both NESC (p < 0.01) and EUSC (p < 0.01) (Figs. 7A–7E).

Figure 6 The mRNA expression levels of Gal-3, COL-1, CTGF, and α-SMA in endometrial stromal cells.

The mRNA expression levels of (A) Gal-3, (B) COL-1, (C) CTGF, and (D) α-SMA were detected by qRT-PCR in tissue samples. Data were presented as mean ± SD. * p < 00.05, ** p < 00.01. NESC, normal endometrial stromal cells; ECSC, ectopic endometrial stromal cells; EUSC, eutopic endometrial stromal cells.

Figure 7 The protein expression levels of Gal-3, COL-1, CTGF, and α-SMA in endometrial stromal cells.

(A) The protein expression levels of (B) Gal-3, (C) COL-1, (D) CTGF, and (E) α-SMA were detected by Western blot analysis in endometrial stromal cells. Data were presented as mean ± SD. * p < 00.05, ** p < 00.01. NESC, normal endometrial stromal cells; ECSC, ectopic endometrial stromal cells; EUSC, eutopic endometrial stromal cells.

Interference efficiencies of shRNAs in immortalized human endometriotic cells

We then knockdowned Gal-3 in 12Z human endometriotic cells through transfecting the constructed shRNAs with different sequences, and the knockdown efficiencies of the constructed shRNAs were verified. The results showed that shRNA-1, 2 and 3 could effectively reduce the Gal-3 mRNA expression level compared with negative control (p < 0.01). However, shRNA-1 was more effective in inhibiting the Gal-3 mRNA expression level than shRNA-2 and shRNA-3 (Fig. 8A). Therefore, we used S1 sequence for subsequent study.

Figure 8 The effect of Gal-3 on the mRNA expression levels of fibrosis-related factors in immortalized human endometriotic cells.

(A) The interference efficiency of Gal-3 shRNAs in Z12 cells. RT-qPCR was used to determine the mRNA expression levels of (B) Gal-3, (C) COL-1, (D) CTGF, and (E) α-SMA in Z12 cells following transfection with Gal-3-OE or Gal-3-shRNA. Data were presented as mean ± SD. * p < 0.05, ** p < 00.01.

The influence of Gal-3 on fibrosis-related factor expression in immortalized human endometriotic cells

To elucidate the effect of Gal-3 on fibrotic changes during the progression of OVE, we conducted gene modulation experiments in 12Z endometrial cells. Specifically, we achieved overexpression and knockdown of Gal-3 through transfection with Gal-3-OE and Gal-3-shRNA constructs, respectively. The efficiencies of these genetic intervention were documented in Fig. 8B. Remarkably, Gal-3-OE transfection led to a significant upregulation of Gal-3 mRNA level compared to the NC-OE group (p < 0.01), while Gal-3-shRNA transfection effectively downregulated Gal-3 mRNA expression in 12Z cells compared to the NC-shRNA group (p < 0.01). Moreover, our results revealed that the mRNA levels of COL-1, CTGF, and α-SMA were substantially upregulated in 12Z cells after Gal-3 overexpression (p < 0.01) (Figs. 8C–8E). Conversely, these gene expression levels were notably diminished following Gal-3 knockdown (p < 0.01) (Figs. 8C–8E). Additionally, it was observed that Gal-3-OE transfection increased Gal-3 protein level compared to the NC-OE transfection (p < 0.01), while Gal-3-shRNA transfection reduced Gal-3 protein level in 12Z cells compared to NC-shRNA group (p < 0.01) (Figs. 9A, 9B). Likewise, Gal-3 overexpression elevated the protein levels of COL-1, CTGF, and α-SMA in 12Z cells (p < 0.01), while Gal-3 knockdown inhibited the protein expression of COL-1, CTGF, and α-SMA (p < 0.01) (Figs. 9A, 9C–9E) Galectin-3. Therefore, our results revealed that Gal-3 promoted fibrosis in human endometriotic cells and is involved in fibrosis development in endometriosis.

Figure 9 The effect of Gal-3 on the protein expression levels of fibrosis-related factors in immortalized human endometriotic cells.

Western blot analysis was used to detect (A) the protein expression levels of (B) Gal-3, (C) COL-1, (D) CTGF, and (E) α-SMA in Z12 cells after transfecting Gal-3-OE or Gal-3-shRNA. Data were presented as mean ± SD. * p < 00.05, ** p < 00.01.

Discussion

Endometriosis, a multifaceted gynecological ailment characterized by the atypical development of endometrial tissue and stroma at locations outside the cavity of the uterus (Nanda et al., 2020). With the progress of endometriosis, pelvic adhesion gradually worsened. Studies have shown that more than 80% of endometriosis patients have adhesion formation during the initial operation, and the incidence of re-adhesion three months after operation can reach 50% (Somigliana & Garcia-Velasco, 2015). The subclinical peritonitis or mild pelvic adhesion of mild endometriosis can develop into severe and extensive adhesion of severe endometriosis, pelvic fibrosis, and even a frozen pelvic cavity (Guo, Du & Liu, 2016). Nonetheless, despite ongoing advancements in biomedical research, the pathology associated with endometriosis is still not fully known (Wang, Nicholes & Shih, 2020). Endometrial fibrosis is a critical pathological condition that plays a substantial role in the onset and advancement of endometriosis. Previous studies have indicated that Gal-3 holds significant potential as both a diagnostic tool and a target for therapeutic interventions in fibrotic disorders in several organ systems, including the cardiac, pulmonary, and renal tissues (Blanda et al., 2020; Fulton et al., 2019; Wu et al., 2022). Nevertheless, the precise extent of participation of the Gal-3 and its molecular function in the development of fibrosis in OVE has not been shown.

The development of fibrosis is induced by abnormal fibroblasts activation and transformation to myofibroblasts (Venugopal et al., 2022). α-SMA is a key marker for myofibroblasts, and the characteristics of activated myofibroblasts is the production of a large quantity of extracellular matrix, especially COL-1 (Dolivo, Weathers & Dominko, 2021). CTGF, a member of the CCN matricellular protein family, is known to control the signaling of growth factors and promote fibrosis. It was found that mice exhibiting increased CTGF in fibroblast are vulnerable to accelerated tissue fibrosis (Toda et al., 2018). Therefore, our study investigated the effects of Gal-3 on fibrosis markers including COL-1, CTGF, and α-SMA in endometriosis. The findings of our study provide clear evidence of increased expression of Gal-3, COL-1, CTGF, and α-SMA in ectopic endometrial tissues of OVE patients compared to the eutopic endometrial tissues of OVE patients and non-OVE controls. Moreover, the expression levels of Gal-3, COL-1, CTGF, and α-SMA in ECSC were all upregulated in comparison to NESC and EUSC. Subsequent mechanistic assays revealed that Gal-3 overexpression potentiates the mRNA and protein expression of COL-1, CTGF, and α-SMA, whereas Gal-3 knockdown engendered the opposite effect. These findings collectively substantiate the contributory role of Gal-3 in the fibrotic pathogenesis associated with endometriosis. This also highlighted the contributory role of inherent alterations in eutopic endometrial tissues in OVE pathogenesis. It was previously delineated that Gal-3 expression varies according to the menstrual-cycle and is preeminently elevated in the ectopic endometrial tissues during proliferative or secretory phases (Noël et al., 2011). Moreover, Caserta et al. (2014) revealed that Gal-3 level was prominently higher in endometriosis group than in controls, and Gal-3 level was positively correlated with the stage of endometriosis. These reported results are similar with our results that Gal-3 was upregulated in ectopic endometrial tissues of OVE patients. Notably, Gal-3 was implicated in fostering cellular proliferation, inflammation, and nutrient uptake in endometriotic lesions, thereby facilitating endometriosis development (Hisrich et al., 2020). Gal-3 deficiency significantly impaired endometriosis development in experimental endometriosis mice. These findings emphasized the momentous role of Gal-3 and its potential as a treating target in endometriosis development.

Prior investigations have suggested that Gal-3 is a key regulator in fibrotic disorders affecting the heart, renal system, and pulmonary tissues, and Gal-3 inhibition may be an effective treatment method for fibrosis diseases (Blanda et al., 2020; Fulton et al., 2019; Wu et al., 2022). Our study postulates a similar mechanism operative in endometrial tissues, and our results proved that Gal-3 overexpression potentiates the expression of fibrosis-related markers, indicating the contributory role of Gal-3 in the fibrotic pathogenesis associated with endometriosis. The available evidence has underscored that elevated Gal-3 expression concomitantly escalates vascular endothelial growth factor (VEGF) and its receptor expression levels, thereby enhancing vascular density and pro-angiogenic factors within the endometriotic lesions in endometriosis patients (Mattos et al., 2019). Furthermore, accumulating evidence points toward a surge in macrophage populations, particularly M2 macrophages, within endometriotic lesions. Gal-3 was secreted predominantly by monocytes and macrophages, which instigates the transformation of quiescent fibroblasts into active myofibroblasts, a hallmark of tissue fibrosis. Concomitant activation of M2 macrophages implicated in fibrotic processes amplifies Gal-3 expression and secretion (Henderson et al., 2008). These macrophages orchestrate the secretion of an array of bioactive molecules, including, transforming growth factor-β (TGF-β) and VEGF. This secretory milieu fosters cellular proliferation and angiogenesis, thereby perpetuating both endometriotic lesion progression and fibrotic transformations (Xiao, Liu & Guo, 2020; Yang & Zhang, 2017). The above findings and our present findings posit that Gal-3 overexpression within endometriotic lesions serves as a nidus for fibrotic induction. Consequently, therapeutic strategies targeting the inhibition of Gal-3 and fibrogenesis emerge as promising avenues for future endometriosis interventions. Our findings provide valuable insights into the molecular mechanisms underlying fibrosis in endometriosis. However, further mechanism investigations are needed to elucidate the precise regulatory pathways of Gal-3 in endometrial fibrosis. Nonetheless, our results indicated that Gal-3 promotes fibrosis in endometriosis and act as a possible therapeutic target, which expands on our knowledge of the pathophysiology of fibrosis in endometriosis and opens avenues for future research in endometriosis.

There are still a few significant potential limitations to this study. The availability and robustness of our results may have been impacted by the small sample size we had access to for OVE patients, as well as confounding or biased factors, such as the small sample size of experimental tissues, and systematic errors caused by experimental methods. Moreover, our findings suggest substantiate the contributory role of Gal-3 in the fibrotic pathogenesis associated with endometriosis. However, the precise regulatory pathways of Gal-3 in endometrial fibrosis have not been elucidated yet. In our future research, we will increase the sample size of experiments and work to eliminate potential confounding or bias factors in order to obtain legitimate research results. Additionally, we intend to explore precise molecular regulatory mechanisms of Gal-3 in endometrial fibrosis.

Conclusion

In summary, our investigation delineates a prominent upregulation of Gal-3 in the ectopic endometrial tissues of OVE patients. Gal-3 overexpression significantly escalates the mRNA and protein levels of key fibrosis-related genes including COL-1, CTGF, and α-SMA, while Gal-3 knockdown showed the opposite results. Our results substantiate the contributory role of Gal-3 in the fibrotic pathogenesis associated with endometriosis, indicating that Gal-3 has the potential to be a promising therapeutic target for treating fibrosis in endometriosis.

Supplemental Information

Supplemental Information 1 Raw Data

Click here for additional data file.

Supplemental Information 2 Western blot images

Click here for additional data file.

Supplemental Information 3 MIQE checklist

Click here for additional data file.

Additional Information and Declarations

Competing Interests

Author Contributions

Human Ethics

Data Availability

The authors declare there are no competing interests.

Guimin Yang conceived and designed the experiments, performed the experiments, prepared figures and/or tables, authored or reviewed drafts of the article, and approved the final draft.

Yupeng Deng conceived and designed the experiments, analyzed the data, prepared figures and/or tables, and approved the final draft.

Guangming Cao performed the experiments, analyzed the data, authored or reviewed drafts of the article, and approved the final draft.

Chongdong Liu conceived and designed the experiments, performed the experiments, analyzed the data, prepared figures and/or tables, authored or reviewed drafts of the article, and approved the final draft.

The following information was supplied relating to ethical approvals (i.e., approving body and any reference numbers):

All samples obtained in this study were approved by the ethics committee of the Beijing Chaoyang Hospital of Capital Medical University and abided by the ethical guidelines of the Declaration of Helsinki.

The following information was supplied regarding data availability:

The raw data is available in the Supplementary File.

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
