# Peer review of "Galectin-3 promotes fibrosis in ovarian endometriosis"

_PeerJ, doi:10.7717/peerj.16922_

## Round 0.1 · original submission · Major Revisions

It requires a number of Major Revisions.

Reviewer 1 ·

Basic reporting

The manuscript presents a study investigating the role of Galectin-3 (Gal-3) in the development of fibrosis in ovarian endometriosis (OVE). The main findings indicate an increased expression of Gal-3 in ectopic endometrial tissues of OVE patients compared to their eutopic counterparts and patients without OVE. Additionally, the expression of fibrosis-related markers COL-1, CTGF, and α-SMA was elevated in ectopic endometrial tissues of OVE patients. The study demonstrates that Gal-3 overexpression enhances the expression of these fibrosis-related markers, while Gal-3 knockdown has the opposite effect.
Regarding the technical standard employed, the study demonstrates detailed experiments involving gene modulation through transfection with Gal-3-overexpressing and Gal-3-short hairpin RNA constructs. The knockdown efficiency of Gal-3 is verified, and the impact on the expression of fibrosis-related markers is assessed through molecular profiling and western blot analyses. The methods appear to be replicable and technically sound.
The availability and robustness of the underlying data are not explicitly discussed in the manuscript. However, the statistical analyses performed, such as the significance testing (p-values) of gene expression and protein levels, contribute to the statistical soundness of the results. The manuscript also lacks information on specific control measures implemented.
Overall, the study provides clear evidence of the involvement of Gal-3 in fibrosis development in OVE. The findings offer valuable insights into the molecular mechanisms underlying endometriosis-associated fibrosis and suggest Gal-3 as a potential therapeutic target. However, further investigations are needed to elucidate the precise regulatory pathways of Gal-3 in endometrial fibrosis and to assess the availability, robustness, and control measures of the underlying data. Nonetheless, the study contributes to our understanding of the pathogenesis of fibrosis in endometriosis and opens avenues for future research in this field.

Experimental design

1. The authors should provide clear information on the sample size and selection criteria for the patients included in the study. This is essential for understanding the generalizability of the findings.
2. Describe the inclusion and exclusion criteria for selecting the patients included in the study. This will help readers understand the characteristics of the sample population.
3. Provide detailed information on the Gal-3 overexpressing and knockdown techniques used in the experiments. Explain the design of the constructs and the specific methods used for transfection.

Validity of the findings

1. The results of the gene modulation experiments in 12Z endometrial cells are presented without sufficient detail. The specific methods and controls used in the transfection experiments should be described in more depth.
2. The discussion section should provide a critical analysis of the findings, including a comparison with previous research and potential implications of the results. Currently, the discussion is limited and does not sufficiently analyze the findings in the broader context of the field.
3. Provide a brief summary of the limitations of the study, including potential sources of bias or confounding, and suggestions for future research to overcome these limitations.
4. Revise the conclusion section to provide a succinct summary of the main findings and their implications. Avoid introducing new information or repeating earlier sections.

Additional comments

1. In the Introduction, it would be helpful to provide a more concise and clear background on endometriosis and its association with fibrosis. The current explanation is lengthy and lacks focus.
2. The language and writing style throughout the manuscript need improvement. Simplify the language and restructure sentences to enhance clarity and readability.
3. Define all relevant abbreviations and ensure consistency in their use throughout the manuscript. This will prevent confusion and aid comprehension.

Cite this review as

Reviewer 2 ·

Basic reporting

The availability, statistical soundness, and control measures of the underlying data are adequately addressed. Overall, this manuscript provides valuable insights into the understanding and potential treatment of fibrosis in OVE.

Experimental design

a) The manuscript would benefit from an introductory paragraph that provides an overview of endometriosis and its relevance to fibrosis. This would provide a strong foundation for the study.
b) Clearly articulate the research questions or hypotheses that motivated the study. This will guide the reader and help establish the purpose of the research.

Validity of the findings

a) In the Results section, the findings should be presented in a clear and concise manner. Currently, the language is convoluted and difficult to follow. Simplify the language to enhance readability.
b) The authors employ abbreviations such as TGF-β and COX-2 without explanation. All abbreviations should be defined at their first mention to ensure clarity for readers.
c) Provide specific examples from the literature to support the statements made in the discussion section. This will strengthen the arguments and enhance the credibility of the findings.
d) Consider the implications of the findings and the potential applications in clinical practice or further research. Discuss how the results may contribute to the understanding and treatment of endometriosis.
e) Consider including a subsection in the discussion that addresses potential limitations of the study, such as sample size, generalizability, or potential biases.
f) In the paragraph discussing the results of protein and mRNA expression levels, clarify the specific target proteins being analyzed (Gal-3, COL-1, CTGF, α-SMA). Provide their relevance to endometriosis.

Additional comments

a) The Introduction should clearly state the objective of the study, including the specific research questions that the authors aim to address. This will help readers understand the purpose of the research.

Cite this review as

Reviewer 3 ·

Basic reporting

.

Experimental design

.

Validity of the findings

.

Additional comments

Abstract

Change the data position about samples collection to before presenting the number of samples collected (line 15);
Gal-3 overexpression – indicate that the transfection is from the vectors; line 19;


Methods

Provide antibodies specifications and dilution (line 109; line 136);
Review the order of the methods presented with the aim of following a timeline of experiments - Hematoxylin and Eosin (H&E) Staining (line 154) Masson staining (line 164);
Western Blotting (line 171) – describe which antibodies were used, antibodies specifications and dilution

Results
Figure 9 – Figure subtitle – Replace “Western blot analysis was used to detect (A) the mRNA expression levels...” with “Western blot analysis was used to detect (A) the protein expression levels...”

Cite this review as

---

## Round 0.2 · accepted · Accept

I am writing to inform you that your manuscript - Galectin-3 promotes fibrosis in ovarian endometriosis - has been Accepted for publication. Congratulations!

Reviewer 1 ·

Basic reporting

I have reviewed response letter and it was well.

Experimental design

I have reviewed response letter and it was well.

Validity of the findings

I have reviewed response letter and it was well.

Additional comments

I have reviewed response letter and it was well.

Cite this review as

Reviewer 2 ·

Basic reporting

authors have addressed questions well

Experimental design

authors have addressed questions well

Validity of the findings

authors have addressed questions well

Additional comments

authors have addressed questions well

Cite this review as